# Photobiomodulation for Chemotherapy-Induced Oral Mucositis in Pediatric Patients

**DOI:** 10.3390/biom13030418

**Published:** 2023-02-23

**Authors:** Daša Hafner, Petra Hrast, Tanja Tomaževič, Janez Jazbec, Marko Kavčič

**Affiliations:** 1Faculty of Medicine, The University of Ljubljana, 1000 Ljubljana, Slovenia; 2Department of Pediatric and Preventive Dentistry, Dental Clinic, University Medical Center Ljubljana, 1000 Ljubljana, Slovenia; 3Department of Pediatric Hematology and Oncology, Division of Pediatrics, University Medical Centre Ljubljana, 1000 Ljubljana, Slovenia

**Keywords:** oral mucositis, oral stomatitis, low-level laser therapy, photobiomodulation, chemotherapy, pediatric population, pediatric oncology

## Abstract

Oral mucositis (OM) is a common side effect in patients undergoing chemotherapy (CT), especially in children due to their rapid epithelial mitotic rate. It has been associated with a significant reduction in life quality since it leads to pain, an inadequate intake of nutrients, an increased risk of opportunistic infections, and interruptions of CT. Photobiomodulation (PMB) with low-level laser therapy (LLLT) has shown faster healing, reduction in pain, and the reduced use of analgesic compared to placebo groups. The purpose of this review is to analyze and compare the existing clinical trials and identify their shortcomings in hope to make future research easier. Using MeSH terms and keywords, the Embase, Medline, and PubMed databases we searched for the period of the last 5 years. We identified a total of 15 clinical trials, with a total of 929 pediatric patients analyzed in this review. We compared different light sources and other laser technique characteristics used in clinical trials such as wavelength, energy and power density, spot size, irradiation time, PBM protocol, and OM evaluation. The main findings show inconsistent laser parameter quotations, differences in the PBM protocol along with a laser application technique, and a lack of clinical trials. Based on that, more studies with a high methodological quality should be conducted in order to provide a unified PBM protocol suitable for the pediatric population.

## 1. Introduction

Oral mucositis (OM) is considered one of the most debilitating complications associated with anticancer therapy [1]. It begins as an inflammation process at the basal epithelium of the oral mucosa, which gradually develops into mucosal erythema and ulcers [2,3]. Intense pain and discomfort related to ulcer formation can lead to impaired oral functions, such as the ability to speak and swallow solid or even liquid food. It puts the patients at risk of malnutrition and increases the need for parenteral nutrition and strong analgetic drugs such as opioids. The impaired function of the epithelial barrier may lead to secondary and systemic infections, which result in higher demands for wide-spectrum antibiotics [1,4,5,6]. Besides that, OM is often associated with cancer therapy interruptions or dose reductions, resulting in worsened clinical and economic outcomes and higher patient mortality [5,6,7].

Due to rapidly dividing cells, oral mucosa is highly sensitive to the damaging effects of cytotoxic drugs or radiotherapy (RT) to the head and neck region. Children and young adults are thus predisposed to develop this condition because they have a higher epithelial mitotic rate [5,7,8]. Age, as well as the type of malignancy and treatment regime, are some factors that influence the incidence of chemotherapy-induced OM. It is the highest amongst individuals who receive head and neck RT (almost 100%), followed by those undergoing high-dose chemotherapy prior to hematopoietic stem cell transplantation (HSCT) (about 80%). In children, OM occurs in approximately 52 to 80% of cancer patients receiving treatment, although there are some variations in the literature. The estimated incidence is higher in patients with hematological malignancies compared to solid tumors [6,7,9,10,11,12]. Furthermore, some drugs used in different chemotherapy (CT) protocols display a higher cytotoxic potential than others, one example being methotrexate (MTX) alone or in combination with other agents. They are first used in the treatment of acute lymphoblastic leukemia (ALL)—the most common childhood malignancy—and conditioning regiments prior to HSCT [12,13,14].

Because of the consequences of OM on the life quality of childhood cancer patients, it is important to try to prevent and treat this condition carefully. There are many strategies that may help to decrease the incidence, severity, and duration of OM, such as oral hygiene protocols, topical anesthetics, analgesics (opioid and non-opioid drugs), antimicrobial agents (chlorhexidine), anti-inflammatory agents (benzydamine), cytoprotective agents (glutamine), growth factors (keratinocyte growth factor (KGF)), cryotherapy, and photobiomodulation (PBM) [5,15,16]. In recent years, PBM, or as it was previously known, low-level laser/light therapy (LLLT), has shown great potential in the prevention as well as treatment of OM. The term is used to describe the usage of laser devices emitting light in the red and near-infrared spectrum at very low, non-thermal doses.

PBM has been found to alleviate pain and inflammation and modulate the immune response while promoting tissue repair and regeneration [15,16]. It does so through a series of photophysical and photochemical reactions, starting with the absorption of laser light energy by endogenic photoreceptors situated in the mitochondrial membrane. This results in the conversion of light photons to a form of energy that cells are able to use in a state of higher metabolic demand. The products of those are involved in modulating gene transcription, blood flow, lymphatic drainage, and peripheral nerve activity. The results, paired with the absence of side effects, feasibility, and a good rate of reducing the incidence, duration, and OM-associated pain, are the reason for the rapid growth of interest in PBM in pediatric oncology [1,10,15,16].

However, a problem that occurs is that there is a lack of robust evidence and high-quality clinical trials to determine the efficacy of different PBM protocols. That prevents the use of PBM in standard clinical practice. Another line of doubt arises from the fact that the evidence is largely based on studies performed on the adult population, while a protocol for the pediatric population has not yet been established. The guidelines for the adult population were proposed by the Multinational Association of Supportive Care in Cancer and the International Society for Oral Oncology (MASCC/ISOO). They suggested the following dosimetric parameters: a wavelength of 632.8 nm, a 31.25 mW/cm^2^ power density, a 1.0 J/cm^2^ energy density with 40 s per spot to be used for patients undergoing HSCT, and a wavelength of 660 nm, a 417 mW/cm^2^ power density, and a 4.2 J/cm^2^ energy density with 10 s per spot for patients treated with a combination of CT and RT to the head and neck [15]. Children’s oral mucosa exerts different biological properties and childhood malignancies are treated with different treatment modalities; therefore, an accordingly adjusted PBM protocol is needed [15,16].

## 2. Methodology

### 2.1. Search Strategy

Our literature review consisted of the search for relevant studies published in the last 5 years (from 2017 to now). Databases such as Embase, Medline, and PubMed have been screened using MeSH terms (‘Stomatitis, Aphthous’, ‘Mucositis’, ‘Phototherapy’, ‘Low-Level Light Therapy’, ‘Lasers, Semiconductor’, ‘Laser Therapy’, ‘Photochemotherapy’, ‘Antineoplastic Combined Chemotherapy Protocols’, ‘Antineoplastic Agents’, ‘Cancer Chemotherapy’, ‘Precursor Cell Lymphoblastic Leukemia-Lymphoma’, ‘Hematopoietic Stem Cell Transplantation’) and other keywords related to PBM and chemotherapy-induced OM in children (‘Oral mucositis’, ‘Chemotherapy-induced Oral Mucositis’, ‘Low-Level Laser therapy’, ‘Photobiomodulation’, ‘LLLT’, ‘Chemotherapy’, ‘Acute Lymphoblastic Leukemia’, ‘Bone Marrow Transplantation’, ‘Pediatric Oncology’, ‘Childhood Cancer’). We aimed to phase out only pediatric clinical studies conducted on children undergoing cancer treatment and therefore at risk of developing OM. Our inclusion criteria were pediatric clinical trials and retrospective studies. The search was limited to literature written in the English language only, and the age of the participants was set to be from 0 to 18 years old (children and adolescents) or either from 19 to 24 years old (young adults). This database search was complemented by the manual search for the most recent systematic reviews and meta-analyses regarding our field of interest; hence, we were also able to screen their references to identify additional studies that were not retrieved in the original search.

### 2.2. Study Selection

The identified records were screened by 2 authors independently (k score was 0.8). We excluded the studies that focused on stomatitis or other etiologies and studies performed on animal models and the adult population. After reading the abstracts and excluding all the irrelevant literature, we were left with 15 studies that were included for the purpose of this review (either randomized controlled trials (RCTs), cohort, retrospective, or feasibility studies) and some of the most recent systematic reviews and meta-analyses. We analyzed the full text of 15 retrieved research papers (Figure 1).

## 3. Results

### 3.1. The Role of Photobiomodulation in the Treatment of Chemotherapy-Induced Oral Mucositis

In total, 4 out of 15 retrieved papers were randomized controlled trials (RCTs) comparing the PBM treatment of OM lesions caused by chemotherapy to no treatment [8,17,18,19]. Two of them were controlled by a placebo treatment and the patients, along with their parents, were blinded to what kind of therapy they have received [17,18]. All four of them had similar primary outcomes; that is, to evaluate whether PBM was effective in decreasing the severity of OM and alleviating pain. The results showed that there was a statistically significant reduction in OM grade as well as a reduction in patient-reported pain in all of the RCTs, even though they used different PBM protocols. The diode laser devices that were used by the authors emitted light with a wavelength of 660 to 970 nm, power from 150 to 1500 mW, and an energy fluency from 4.5 up to 26.8 J/cm^2^. There were also slight differences in the light application technique and the schedule of PBM treatments (Table 1).

### 3.2. The Efficacy of Photobiomodulation Compared to Other Treatment Modalities for Treating Chemotherapy-Induced Oral Mucositis

The next group of identified studies was three RCTs that compared PBM therapy to other treatment options, such as photodynamic therapy (PDT), [20,21] and used LED devices instead of diode lasers as light sources [22]. PDT is the application of a substance that acts as a photosensitizer (in our case, methylene blue), over the tissue surface a few minutes prior to laser light treatment. This method aids in overcoming oral infections due to its antimicrobial and antifungal properties. Irradiated with laser light and in the presence of oxygen, photosensitizing substances have the ability to form reactive oxygen species (ROS), which then act selectively on oral microbiota, resulting in their death [20]. Since PBM is already considered effective in treating OM, the authors in both studies hypothesized that PDT is able to enhance the effectiveness of PBM. The first study had a split-mouth design, where one side of the oral cavity was treated with PDT and the other side was treated with PBM alone. Patients were blinded to the treatment modality. They observed a significantly smaller lesion on the PDT-treated side in comparison to the side treated solely with PBM [20]. In the other RCT, the authors did not report any significant difference in the OM severity or OM-related pain; thus, both treatment options were equally effective [21]. A similar result was observed with LED as a light source compared to diode lasers. There was no difference regarding the OM grade nor OM-associated pain [22]. There were again slight differences in the PBM protocols (Table 2).

### 3.3. The Role of Photobiomodulation in the Prevention of Chemotherapy-induced Oral Mucositis

In recent years, PBM has been used not only to treat OM, but also to prevent it. We retrieved three retrospective studies that focused on the prevention of OM rather than the treatment of it solely [7,23,24]. All the pediatric cancer patients underwent prophylactic PBM protocol, and only those who developed OM were treated with curative PBM protocol accordingly [23,24]. Those studies aimed to report the incidence of OM and OM-associated clinical outcomes in correlation with prophylactic PBM to decipher if a laser treatment would also be useful in the prevention of OM. The authors came to some interesting conclusions. They observed that OM incidence corresponds with a primary cancer diagnosis and treatment options. A higher occurrence of OM was observed in osteosarcoma, ALL, and HSCT patients, as well as in patients who were treated with MTX [7,23,24]. Again, they agreed on the fact that PBM is capable of preventing OM since patients who underwent prophylactic PBM were less likely to develop the condition, and the reported OM occurrence was less compared to the literature [7,23]. Unfortunately, the data were retrieved retrospectively, and the studies did not have a control group, which decreased the quality of the results (Table 3). To illustrate quality research on the efficacy of prophylactic PBM conducted before our timeframe, we added a systematic review carried out by He et al. to Table 3. By selecting eight clinical trials with a high methodological quality, they were able to demonstrate that prophylactic PBM indeed reduces OM and severe OM in pediatric cancer patients undergoing chemotherapy [6].

### 3.4. In the Search for Optimal Photobiomodulation Protocol

In this review, we also wanted to focus on different PBM protocols since the exact and most beneficial dosimetric parameters for children have not yet been established. A lot of clinical trials use different light sources with different settings and, therefore, dosimetric parameters. They are summarized in Table 4.

Amongst the retrieved articles, we identified two where the authors wanted to reduce the confusion around finding the most effective and patient-friendly PBM protocol. They agreed on the fact that by increasing the energy density and shortening the irradiation time, we can implement shorter PBM protocols. Therefore, energy levels delivered to the oral mucosa stayed the same, which would be more practical, especially for young children. Moreover, it is also sensible to reduce the energy density in correlation to decreasing the OM grade [25]. Another feasibility study that we encountered confirmed the fact that PBM is indeed a feasible and well-tolerated form of OM treatment in children, and another cross-sectional study provided some information on the demographic and clinical characteristics of pediatric patients who have developed OM as a side effect of cancer treatment [4].

Due to the usage of different laser devices, there was a difference in handpieces and thus the surface of the light beam or spot size. The authors used wavelengths ranging from 660 to 980 nm; some of them used a light source that emitted two different wavelengths simultaneously [13,20,25]. One study used different wavelengths according to OM grades on the World Health Organization (WHO) scale; they used the longer one for higher OM grades [4]. The range of energy density varied from 2 to 107 J/cm^2^ and the power ranged from 0.005 to 3.05 W. We encountered two studies that used different energy densities according to the OM grade; they used a higher energy density for a higher OM grade, respectively [25,26].

There is also inconsistency in using different light application techniques (Table 1, Table 2 and Table 3) where some operators used punctual modality [4,7,21,23,24], which means holding the handpiece of a laser device perpendicular to the tissue surface for a certain amount of time, or rather a defocused modality [17,18], where they moved it in circular motions or brushing strokes over a particular surface area. Some operators provided laser therapy in contact with the oral mucosa [20,21,22,23], while others kept some distance between the tissue surface and the tip of the handpiece [17,20,24,26].

Differences also arose from the frequency of laser therapy sessions, where in some cases they were delivered daily [8,8,17,20,21,22], and in other they were delivered every second day [13,19,25,26]. Additionally, some protocols had a definite number of sessions [8,13,17,18,19,20], while others provided PBM treatment until the clinical resolution of the lesion [21,22]. Discrepancies were acknowledged also between the authors’ PBM protocols depending on whether it was used for OM prophylaxis rather than treatment [24]. One study also reported having a different PBM protocol for intraoral and extraoral light application [13].

All of the above can have a detrimental consequence on the amount of energy that we deliver to the damaged oral mucosa, which is the key element in the effectiveness of laser light therapy. Furthermore, a whole new problem arises from inconsistency in reporting all the components of a PBM protocol, such as the description of a light source and all the accompanying dosimetric parameters, light application technique, and treatment frequency.
biomolecules-13-00418-t004_Table 4Table 4Summary of dosimetric parameters used in different clinical trials.First Author, Year, (Reference)UsageLight SourceWavelength (nm)Power (W)Power Density (W/cm^2^)Energy Density (J/cm^2^)Energy Per Point (J)Spot Size (cm^2^)Time (s)Gobbo et al. (2017) [17]TreatmentGaAlAs (diode)660 and 970combined3.232026.88125 per spotVitale et al. (2017) [18]TreatmentGaAlAs (diode)9703.2///1320 per sessionKaraman et al. (2022) [19]TreatmentGaAlAs (diode)8300.15/4.5/130 per spotReyad et al. (2022) [8]TreatmentDiode9801.5/4.5/
30 per spotMadeiros Filho et al. (2017) [20]TreatmentAsGaAl and InGaP (diode)660 and 8080.1/4//90 or 10 per areaRibeiro da Silva et al. (2018) [21]TreatmentInGaAlP(diode)6600.1
35 or 105
0.02830 or 10 per spotNoirrit-Esclassan et al. (2019) [26]Extraoral TreatmentDiode635 and 815combined0.15 and 3.854///from 30 per 30 cm^2^ area to 50 for 50 cm^2^Intraoral TreatmentDiode635 and 815combined0.156///30 per 2 cm^2^
Tomaževič et al. (2019) [27]TreatmentDiode8100.25/8.8 or 15.5 (depending on OM ^1^ grade)//4.4 or 7.8 per spot (depending on OM grade)TreatmentDiode8100.5/8.8 or 15.5 (depending on OM grade)//2.2 or 3.9 per spot (depending on OM grade) TreatmentDiode8100.25/4.4 or 7.75 (depending on OM grade)//2.2 or 3.9 per spot (depending on OM grade)Curra et al. [13]TreatmentInGaAlPDiode6600.1 or 0.04/6///Fiwek et al. [25]TreatmentDiode635 and 980combined1/30//30 per spotTreatmentDiode635 and 980combined0.1/2, 4, 8, or 16 (depending on OM grade)//20, 40, 80, or 160 per spot (depending on OM grade)Cavalcanti et al. (2022) [4]TreatmentGaAlAs and InGaAlP660 or 808(depending on OM grade)0.1/3.3/0.0310 s per spotAvila-Sanchez et al. (2017) [7]Prevention and TreatmentDiode9800.3/18///Nunes et al. (2020) [23]PreventionGaAIAs InGaAIP (diode)66013.3366.620.0320 per spotMiranda-Silva et al. (2021) [24]PreventionDiode6600.1/35.710.02810 per spotTreatmentDiode6600.1/35.720.02820 per spotGuimaraes et al. (2021) [22]Treatment and PreventionInGaAlP (diode)6600.1/20.60.0336 per areaTreatment and PreventionLED6600.005/20.60.785120 per area^1^ Oral mucositis.


## 4. Discussion

### 4.1. Oral Mucositis—Pathobiology and Clinical Assessment

Because OM is such a debilitating and bothersome condition, a lot of research has been made in the last decade to try to understand it better and develop new and effective treatment options. We used to view OM as a simple inflammation process starting in the oral and gastrointestinal mucosa. The reason for its onset was believed to be in the non-specific action of cytotoxic drugs used in anticancer therapies. Now we know that it is a complex sequence of biological events that connect multiple cell types in all layers of mucosal tissue. The immune system and oral microbiota, as well as some environmental factors, also play a crucial part in its pathogenesis [2,3,25]. 

Sonis demonstrated a five-phase model for illustrating pathobiological mucositis development, the first being the initiation phase. It arises soon after the beginning of chemotherapy treatment when cytotoxic agents reach the oral mucosa. There they act directly on the cells and blood vessels under mucosal tissue, making their way to the basal epithelium, which is the next to be affected by their stomatotoxicity. The breakage of DNA strands caused by CT drugs is the reason for the impaired regeneration ability of the oral mucosal tissue. Moreover, damage to the DNA also causes a formation of ROS, which causes further damage and activates multiple signaling pathways and transcription factors. Despite the fact that the onset of OM in the basal layers of the oral mucosa has already begun, the surface seems clinically unchanged at this point [3,28,29].

The formation of ROS leads to primary response phase, which is characterized by excessive activity in intercellular and intracellular signaling pathways. They are able to activate multiple transcription factors, the most common being the nuclear factor kappa-B (NF-κB). This leads to the expression of genes and the formation of pro-inflammatory cytokines, such as interleukin 1 (IL-1), interleukin 1β (IL-1β), and tumor necrosis factor-alpha (TNF-α) [28,29].

The next phase is signaling amplification. It consists of a series of positive feedback loops that serve to amplify and prolong tissue damage through their effect on signaling pathways. Consequently, the production of pro-inflammatory cytokines is increased even further. The tissue of oral mucosa is severely biologically altered, although it may still appear normal to the eye [28,29].

The most clinically significant phase of OM is ulceration, where the mucosal barrier finally loses its integrity. Hence, the well-innervated underlying tissue is exposed, which leads to pain and discomfort. Bacterial colonization of the ulcerations also occurs in this phase, causing even further damage to the oral mucosa. Bacterial cell wall products can activate tissue macrophages, leading to the additional production of TNF-α and other pro-inflammatory cytokines. This presents a high risk of bacteremia and sepsis, especially in patients that experience OM as well as leucopenia caused by CT treatment [2,28,29].

The renewal of epithelial cell proliferation and differentiation are characteristics of the last healing phase of OM. In this phase, which usually lasts from 2 to 4 weeks, the oral mucosal barrier is reestablished and local micro bacterial flora is restored [29]. The pathobiology of OM is represented in Figure 2.

The first clinical presentation of OM is therefore soreness and redness of the oral mucosa. The onset of the first symptoms usually takes place 3 to 4 days after the beginning of CT, and the ulcerations appear towards the end of the first to the second week of anticancer treatment [3,28]. The most affected sites of the oral cavity appear to be less keratinized surfaces of the buccal and lingual mucosa, along with the soft palate and inner aspect of the lips. Sometimes, even oropharyngeal mucosa is affected [1,4,28]. Once multiple ulcers have formed, patients experience extreme pain, which can lead to difficulty speaking and swallowing food as well as liquids. Consequently, there is a higher demand for parenteral nutrition, opioid, and non-opioid drugs, and broad-spectrum antibiotics to treat systemic infections with an origin in the oral cavity. Due to the long list of possible complications, patients are at risk of treatment interruptions, dose reductions, and longer hospital stays, which later result in higher treatment costs and worse clinical outcomes [1,2,5,6]. Furthermore, Bezinelli et al. demonstrated that the presence of severe OM in patients undergoing HSCT is directly linked to a significant decrease in overall survival [26].

There are a lot of different clinical scales used by healthcare providers to assess the severity of OM, which can be the first obstacle in trying to compare different clinical trials in order to evaluate their findings. Most authors use the World Health Organization Oral Mucositis Scale or National Cancer Institute—Common Toxicity Criteria (NCT-CTC). Fortunately, both of them use a five-point scale ranging from 0 to 4, where 0 stands for no symptoms, 1 for mild erythema, 2 for the presence of ulcers with the ability of oral food intake, and 3 and 4 are considered severe forms of OM with impaired oral nutrition intake and parenteral hydration or nutritional support. Pain is usually assessed with a visual analogue scale (VAS), where patients describe their pain with numbers from 0—no pain to 10—worst possible pain. The quality pain assessment can also be challenging sometimes, especially when assessing young children or children that have received analgesia [29,30,31].

### 4.2. Photobiomodulation—Mechanism of Action and Perspectives

PBM is a promising procedure that could be used in the treatment as well as prevention of chemotherapy-induced OM. It applies to the use of red (660–700 nm) or near-infrared light (700–980 nm) produced by a light-emitting device, with the purpose of accelerating tissue repair and regeneration [1,25]. It does so by stimulating different photobiological and photochemical processes on a cell level, without exerting any thermal or damaging effect to the tissue [1,32]. The mechanism of action is based on the conversion of laser light to a form of energy, which fuels the cellular metabolic activity, leading to increased cellular proliferation and protein synthesis. Photoreceptors such as cytochrome c oxidase, situated in the mitochondrial membrane, are able to absorb certain wavelengths of light, which then causes a cascade reaction in the respiratory chain. The end result of this reaction is the production of adenosine triphosphate molecules (ATP) [1,33,34].

Some studies have shown that PBM is able to control and downregulate oxidative stress, which plays a key role in OM pathogenesis. Laser light is able to reduce pro-inflammatory cytokines such as TNF-α, IL-1, and IL-6 and has a blocking effect on NF-kB activity, upregulating anti-inflammatory cytokines [35,36,37]. By that, it increases the production of vascular and other growth factors, resulting in accelerated cell differentiation and migration [32,36]. Besides ATP, nitric oxide (NO) is another byproduct of interactions between light photons and cytochrome c oxidase. It acts as a vasodilator, increasing blood flow to the damaged area, thus allowing more immune cells and oxygen molecules to accumulate in the compromised mucosa. NO has a similar effect on lymph vessels as well; it helps to reduce swelling by increasing the lymphatic flow [1,32,33].

Another aspect of OM is the presence of oral bacterial flora and the inevitable infection of the ulcerations. PBM, or rather photodynamic therapy, may play a role not only in stimulating the local immune system and increasing phagocytotic ability, but it also exerts some anti-bacterial activity. The role of oral microbiome in OM pathogenesis and healing is still unexplored, but is a great prompt for future research [1,20,21,35].

PBM not only accelerates tissue repair and reduces inflammation but it is also able to alleviate pain. Its analgesic properties derive from directly blocking peripheral nervous system functions. By inhibiting the translation of pain signals and reducing action potentials, PBM contributes to decreasing the amount of neurogenic inflammation. Hence, it can provide a non-drug alternative to pain management [38,39].

A problem worth mentioning here is dosimetry and differences between PBM protocols, which can make the results of seemingly similar studies difficult to compare. One of the first parameters that we encounter is the wavelength. The literature suggests that safe and effective wavelengths range from 600 to 1070 nm, but the highest biological usefulness is limited to a spectrum of near-infrared wavelengths that are most likely to be absorbed by chromophores, e.g., cytochrome c oxidase (812 to 846 nm) [1,40]. However, studies using wavelengths out of the optimal window have also demonstrated positive results, hence the efficacy of the PBM treatment is not solely dependent on one parameter [6,10,12,41]. Some authors suggest that different wavelengths could be used for different treatment modalities. Longer wavelengths can penetrate further into the tissue than shorter wavelengths, so they could be useful in treating deeper lesions or an extraoral PBM application [33,40]. This was successfully demonstrated by Noirrit-Esclassan et al., who used a combination of two different wavelengths and two different PBM protocols. One of them was used to deliver light therapy intraorally covering the oral mucosal surface, and the other one was used for extraoral PBM to treat oropharyngeal mucosa, which cannot be easily accessed through the oral cavity [13]. The extraoral application of light can be especially beneficial for non-cooperative pediatric patients [13,40,42]. Using longer wavelengths has the potential to optimize the PBM protocol. Because there is a lower amount of scattering, it allows for an increased dose delivery and thus a shorter treatment duration [40].

The most common light sources used by most health care providers are diode laser devices, emitting coherent light of different wavelengths. Non-coherent light sources such as LED laser devices have already been used in the past and have proven to be successful in preventing and treating OM in children [43,44]. One recent study conducted by Guimaraes et al. has shown that compared to regular laser light, the use of LEDs is just as beneficial [22].

Unfortunately, there are many inconsistencies in the literature regarding the optimal energy dose for managing OM. The interaction between soft tissue and laser light energy is very complex. It is often portrayed as a biphasic dose–response relationship, meaning that when the delivered energy dose is too high or the irradiation time is too long, we will achieve a phototoxic response with cellular inhibition, rather than stimulation. The same applied for the other way around; if laser light exposure is too short or the energy dose is too low, there will be no biological effect at all [1,40,45]. Most authors recommend the energy density (or fluence) to be around 4 or 5 J/cm^2^ with a low irradiance (or power density) of between 10 and 150 mW/cm^2^, but higher as well as lower values are being used in correlation with a different exposure time [1,40,41].

In the light of this complex dose–response phenomenon, Cronshaw et al. suggested that lower energy densities (2–5 J/cm^2^) should be used for OM prophylaxis and lesion healing and that the use of higher energy densities (e.g., 10–15 J/cm^2^) demonstrates a better effect in OM-related pain relief rather than biostimulation [1]. This could be the reason for some controversial study results and an explanation for why it is so difficult to find the optimal PBM protocol parameters. On the other hand, there are multiple clinical trials that observed similar positive effects on OM resolution and pain management despite using different PBM parameters [8,17,18,19,32,46,47,48]. It is therefore sensible to adjust them in a way to make the protocol feasible and child-friendly. Tomaževič et al. did that by demonstrating that it is possible to achieve the same results with a broader spectrum of laser settings, where the most patient-practical setting was the one with a shorter irradiation time [27].

Another thing worth considering is the light application technique, which can have an effect on the amount of energy that is delivered to the tissue. The use of collimated non-contact handpieces is advisable, and larger spot sizes should be used for OM prophylaxis because they can cover a larger mucosal area. Smaller spot sizes are suitable for targeted treatment [1,12]. 

Usually, interventions that are effective in the adult population have a similar effect on children. When it comes to PBM, factors such as age-related variances and lack of patient cooperation can lead to lower procedure success rates. Only a few systematic reviews have addressed this issue, and large well-controlled studies on the pediatric population exclusively are lacking to determine the exact dose and protocol recommendations [5,30]. With our literature search, we were able to retrieve some high-quality studies that focused on the use of PBM in the treatment of chemotherapy-induced OM in children, which rendered similar clinical outcomes, and further elaborated the importance of laser therapy when it comes to OM healing and pain management [8,17,18,19]. Unfortunately, studies regarding OM prophylaxis were of a lower quality. However, some older studies have suggested the positive effect of prophylactic PBM on OM incidence and associated complications [47,48,49]. The rationale behind the idea of prophylactic PBM is to prepare the oral mucosal tissue for oxidative stress produced by anticancer therapies, and therefore increase its capacity to survive. It is also noteworthy to mention that the destructive cytotoxic process begins soon after CT drugs reach the oral mucosa. After that, it takes several days for us to notice the changes on the surface of the oral cavity and for the very first OM symptoms to present [1,28,29].

Some studies indicate that children may be at higher risk of developing OM than adults. They have a higher percentage of rapidly dividing cells and quicker cell turnover. When receiving the same treatment, OM incidence will be higher in the pediatric rather than the adult population. Children are also more frequently exposed to CT drugs, which are associated with higher stomatotoxicity [8,17,46]. One example is the use of MTX, especially in high doses during the consolidation phase of CT protocol for ALL. It is also used for the treatment of some other common childhood malignancies such as osteosarcoma and brain tumors. Curra et al. demonstrated that children exposed to high-dose MTX alone or in combination with cyclophosphamide and doxorubicin experienced a higher incidence and severity of OM. A similar observation was made by Saramento et al. where the use of MTX and melphalan as a part of HSCT conditioning was associated with a higher incidence of OM [14,27]. Different CT protocols used for the treatment of childhood malignancies consist of drugs with different stomatotoxic potential, then this is something that needs to be taken into consideration while designing new clinical trials regarding OM prophylaxis. Studies will yield better and more comparable results if authors will compare children undergoing the same cancer treatment, which correlates to a similar risk of OM development [22,23,24,32,44].

To this day, the Multinational Association of Supportive Care in Cancer and the International Society for Oral Oncology (MASCC/ISOO) provided clinical practice guidelines with the exact PBM protocol and dosimetric parameters for the prevention of OM in adult cancer patients receiving HSCT, RT, or radio chemotherapy to the head and neck region. For childhood cancer patients the panel was, unfortunately, unable to recommend the exact PBM settings due to conflicting results and diverse protocols [15,16]. The Mucositis Prevention Guideline Development Group recommends PBM therapy to pediatric patients undergoing autologous or allogeneic HSCT or patients receiving radiotherapy for head and neck malignancies. They were able to specify one of the PBM protocol parameters, which is wavelength in the red-light spectrum (620–750 nm) [50].

## 5. Conclusions

With this review, we wanted to shed some light on the challenges regarding research on the PBM used for managing OM, which is a common, debilitating, and sometimes life-threatening side effect of anticancer treatment, especially in children. PBM is already an established method for OM prevention and treatment in the adult population, with strong evidence supporting the recommended protocols. It is a safe, feasible, and effective method to perform in pediatric and young patients, but further research to investigate the optimal dosimetric parameters should be performed. To do that, device settings and PBM protocols should be listed carefully, and studies of a high methodological quality should be designed.

As stated before, there have been some recommendations of protocol for adults, but it is unfortunately still unclear how they should be modified for children in order to obtain the best results. It has already been stated that it is reasonable to use a shorter irradiation time at the expense of using a higher energy density and power for the same effect, thus making the PBM protocol more child-friendly. Further research prompts that we wish to propose are the exploration of different wavelength combinations in comparison to using only one wavelength, which wavelengths are the most beneficial, whether daily treatment is more effective than receiving PBM sessions every other day, and, of course, studies that would assess other laser device parameters such as power, energy density, the light application technique, and last but not least, to investigate the perspectives of extraoral PBM.

## Figures and Tables

**Figure 1 biomolecules-13-00418-f001:**
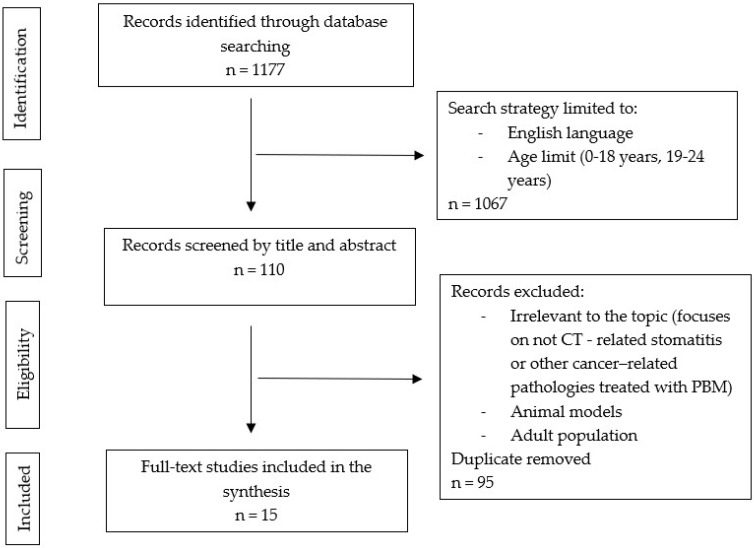
Visual representation of clinical study selection.

**Figure 2 biomolecules-13-00418-f002:**
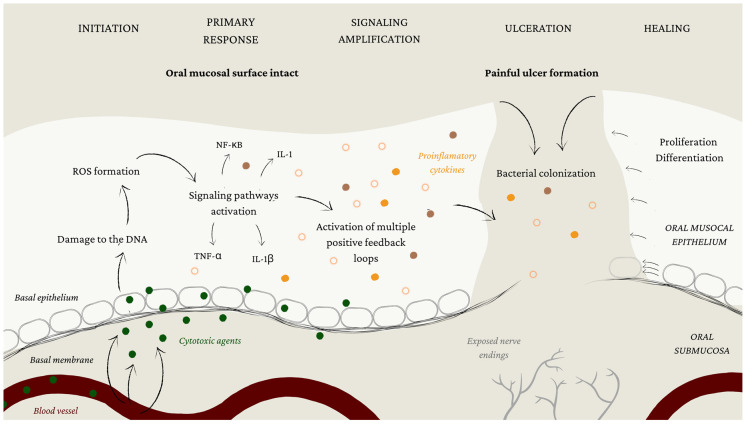
Oral mucositis pathobiology.

**Table 1 biomolecules-13-00418-t001:** Studies addressing PBM treatment of chemotherapy-induced OM in children compared to placebo or no treatment.

First Author, Year, (Reference)	Participants (Number)	Underlying Diagnosis	PBM ^4^ Protocol	Light Application Technique	Treatment Frequency	Outcomes for OM ^5^ Severity	Outcomes for Oral Pain Reduction
Gobbo et al. (2017) [17]	101 51 laser group, 50 placebo control group	ALL ^1^, HSCT ^2^, Lymphoma, AML ^3^, solid tumors	Diode Laser, 660 and 970 nm, 320 mW/cm^2^, 26.8 J/cm^2^, spot size 1 cm^2^, 25 s per spot, 9 spots total	Irradiation of the entire oral cavity, defocused, non-contact modality	Daily, 4 consecutive days	Significantly lower OM grade on day 7 in the laser group	Significant reduction in pain on day 7 in the laser group
Vitale et al. (2017) [18]	16 8 laser group, 8 placebo control group	Not specified	GaAlAs Laser, 970 nm, 3.2 W, spot size 1 cm^2^, 320 s per session	Irradiation of the entire oral cavity, defocused modality	Daily, 4 consecutive days	Significantly lower OM grade on day 7 in the laser group	Significant reduction in pain on day 3 in the laser group
Karaman et al. (2022) [19]	40 20 laser group, 20 control group	Leukemia	GaAlAs Laser, 830 nm, 150 mW, 4.5 J/cm^2^, spot size 1 cm^2^, 30 s per spot	Irradiation of affected areas only	Every other day, 3 times in total	Significantly lower OM grade on days 3, 5, 6, and 7 in the laser group	Significant reduction in pain in the laser group
Reyad et al. (2022) [8]	44 22 laser group, 22 control group	ALL	Diode Laser, 980 nm, 1.5 W, 4.5 J/cm^2^, 30 s per spot	Irradiation of affected areas only, non-contact modality	Daily, 4 consecutive days	Significantly lower OM grade on day 14 in the laser group	Significant reduction in pain on day 10 in the laser group

^1^ Acute lymphoblastic leukemia, ^2^ hematopoietic stem cell transplantation, ^3^ acute myeloid leukemia, ^4^ photobiomodulation, ^5^ oral mucositis.

**Table 2 biomolecules-13-00418-t002:** Studies addressing other treatment modalities in comparison to PBM for treatment of chemotherapy-induced OM in children.

First Author, Year, (Reference)	Participants (Number)	Underlying Diagnosis	PBM Protocol	Light Application Technique	Treatment Frequency	Outcomes for OM ^4^ Severity	Outcomes for Oral Pain Reduction
Madeiros Filho et al. (2017) [20]	15 PDT ^1^ on one side of the oral cavity, PBM ^2^ on the other	Leukemia, osteosarcoma, lymphoma, sarcoma, medulloblastoma	InGaP Laser, 660 nm PDT and AsGaAl Laser, 808 nm for PBM, 100 mW, 90 s per site for PDT and 10 s per site for PBM	Irradiation of affected areas only, punctual, contact modality	Daily, 8 consecutive days	Significantly smaller lesions on days 6 to 8 on the side treated with PDT	Not evaluated
Ribeiro da Silva et al. (2018) [21]	29 14 PDT group, 15 PBM group	ALL ^3^, Non-Hodgkin lymphoma, osteosarcoma	InGaAlP Laser, 660 nm, 100 mW, 107 J/cm^2^ PDT group and 35 J/cm^2^ PBM group, spot size 0.028 cm^2^, 30 s per spot PDT group and 10 s per spot PBM group	Irradiation of affected areas only, punctual, non–contact modality	Daily, until OM healing	No significant difference in the number of sessions required to heal OM between the two groups	No significant difference in the reduction in pain between the two groups
Guimaraes et al. (2021) [22]	80 40 LED group, 40 PBM group	ALL	InGaAlP Laser, 660 nm, 100 mW, 2 J/cm^2^, spot size 0.03 cm^2^, 36 s per area LED, 660 nm, 5 mW, 2 J/cm^2^, spot size 0.785 cm^2^, 120 s per area	Irradiation of the entire oral cavity, contact modality	Daily, until OM healing or hospital discharge	No significant difference in OM incidence and grade between the two groups	No significant difference in the reduction in pain between the two groups

^1^ Photodynamic therapy, ^2^ photobiomodulation, ^3^ acute lymphoblastic leukemia, ^4^ oral mucositis.

**Table 3 biomolecules-13-00418-t003:** Studies addressing PBM in the prevention of chemotherapy-induced OM in children.

First Author, Year, (Reference)	Participants (Number)	Underlying Diagnosis	PBM ^5^ Protocol	Light Application Technique	Findings
Avila-Sanchez et al. (2017) [7]	157	Hematological malignancies, solid tumors, CNS ^1^ tumors	Diode Laser, 980 nm, 300 mW, 18 J/cm^2^	Punctual modality	OM occurrence 21.6%, 94% of OM ^6^ episodes were grade 1 or 2, 4% were grade 3, and 2% were grade 4 on the WHO scale. A higher OM occurrence was associated with ALL and osteosarcoma.
Nunes et al. (2020) [23]	148	ALL ^2^, Osteosarcoma, AML ^3^, HSCT ^4^, Burkitt lymphoma	InGaAIP, 660 nm, 100 mW, 3.33 W/cm^2^, 66.6 J/cm^2^, 20 s per point	Punctual, contact modality	In patients who underwent prophylactic PBM, a lower grade of OM was observed. Higher OM occurrence was observed in HSCT and osteosarcoma patients and those receiving MTX ^7^.
Miranda-Silva et al. (2021) [24]	49	HSCT	Diode Laser, 660 nm, 100 mW, 3.57 W/cm^2^, 35.7 J/cm^2^, spot size 0.028 cm^2^, 10 s per point for prevention and 20 s per point for treatment	Irradiation of the entire oral cavity, punctual, non–contact modality	OM occurrence 73.5%; of that, 36.1% were grades 3 or 4 on the WHO ^8^ scale. The diagnosis of ALL and the use of a myeloablative regiment were associated with OM.
He et al. (2017) [6]	373	Childhood hematological malignancies and solid tumors	LED or diode lasers, 660–970 nm, 3 mW–3.2 W, 4–72 J/cm^2^	Different variety of light application	The odds ratio for developing OM and severe OM (grade 3 or 4 on the WHO scale) after prophylactic PBM is significantly lower compared to placebo (*p* = 0.01 and *p* = 0.03).

^1^ Central nervous system, ^2^ acute lymphoblastic leukemia, ^3^ acute myeloid leukemia, ^4^ hematopoietic stem cell transplantation, ^5^ photobiomodulation, ^6^ oral mucositis, ^7^ methotrexate, ^8^ World Health Organization.

## Data Availability

No new data were created or analyzed in this study. Data sharing is not applicable to this article.

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
