# Peer review of "Photobiomodulation for Chemotherapy-Induced Oral Mucositis in Pediatric Patients"

_biomolecules, 2023, doi:10.3390/biom13030418_

Round 1

Reviewer 1 Report

The topic of this review is very interesting, however, the review should be repeated according to the PRISMA. Also the discussion, should be rewritten, for example, the mechanisms of action should be discussed in the first part of discussion and not at the end!

Line 24 page 1 what are the conclusions?

Line 67 page 2 insert reference

Line 81…why have you put “?”?

Line 84….why did you excluded studies before 2017?

Line 85 add the MEshterms

Line 87 you should specify what type of article you have included in your screening (inclusion criteria)?

Please add the number of total searched articles, the number of exclusions (flow chart)

Have you followed the the PRISMA (Preferred Reporting Items for Systematic Review and Meta-Analyses)

statement (Liberati et al., 2009; Moher, Liberati, Tetzlaff, & Altman, 2009)?

Who didi performed the screening of title and abstract? The reviewers were calibrated? Please add the k-score of agreement

Please specify for each study how many light-irradiations were performed for each time (and also how many seconds)

Line 287 PBM can be performed also with LED, not only laser

Line 313 check the sentence

Reviewer 2 Report

1. Please add more number of recent references

2. Data is represented in tabular form that is excellent. If possible one figure related to pathology or targets

Reviewer 3 Report

Please see attached pdf file.

Round 2

Reviewer 1 Report

line 455 " To do that, laser device settings and PBM" In the text usually, the authors refer to "laser" as the unique device able to perform photobiomodulation...on the contrary, as previously stated other devices, like LED can be used as light-source.

This review does not follow PRISMA; mesh terms have not been inserted, and the k-score of calibration has been considered irrelevant by the author... Based on these conclusions, I cannot consider this review of adequate quality.

Reviewer 3 Report

The authors have addressed some of the comments from Reviewers 1 and 2, from different aspects in different sections. Unfortunately, the authors inserted wordy sentences that made the paragraphs too long. As you can see in the revised version of the manuscript, these paragraphs should be rewritten and shortened:

·        Last paragraph of the introduction has 22 lines.

·        Section 3.4. has 31 lines.

·        In line 227, the sentence is cut by mistake “of energy that we”.

·        Some paragraphs in the Discussion section are too long (more than 20 lines). They should be shortened.
